# DRMIME: Differentiable Mutual Information and Matrix Exponential for Multi-Resolution Image Registration

**Abhishek Nan**[1]                                                          ANAN1@UALBERTA.CA
**Matthew Tennant**[2]                                                    MTENNANT@UALBERTA.CA
**Uriel Rubin**[3]                                                              URIELRUBIN@GMAIL.COM
**Nilanjan Ray**[1]                                                             NRAY1@UALBERTA.CA

[1] *Department of Computing Science, Univeristy of Alberta, Edmonton, Alberta, Canada*

[2] *Department of Ophthalmology, University of Alberta, Edmonton, Alberta, Canada*

[3] *Department of Ophthalmology, Hospital Aleman, Buenos Aires, Argentina*

## Abstract

We present a novel unsupervised image registration algorithm using mutual information (MI). It is differentiable end-to-end and can be used for both multi-modal and mono-modal registration. The novelty here is that rather than using traditional ways of approximating MI which are often histogram based, we use a neural estimator called MINE and supplement it with matrix exponential for transformation matrix computation. The introduction of MINE tackles some of the drawbacks of histogram based MI computation and matrix exponential makes the optimization process smoother. Our use of multi-resolution objective function expedites the optimization process and leads to improved results as compared to the standard algorithms available out-of-the-box in state-of-the-art image registration toolboxes empirically demonstrated on publicly available datasets.

**Keywords:** Image registration, mutual information, neural networks, differentiable programming, end-to-end optimization.

## 1. Introduction

Image registration is a common task required for digital imaging related fields that involves aligning two (or more) images of the same objects or scene. In medical image processing, we may wish to perform an analysis of a particular body part over a period of time. Images captured over time, of the same body part or location will change due to changes in the target organ over time as well variability in angle and distance of the target organ from the capture device. Furthermore, different imaging modalities can provide different and additive information for the clinician or researcher regarding human tissue. A common way to perform a holistic analysis is to combine the (complimentary) information from these different modalities. Alignment of the different modalities requires multi-modal registration.

One of the most successful metrics used for multi-modal medical image registration is mutual information (MI) (Pluim et al., 2003), for which the most common computation method is histogram-based. As a result, MI suffers from the curse of dimensionality when multi-channel images are used. Recent MI estimation method MINE (mutual information neural estimation) (Belghazi et al., 2018) offers a way to curb this difficulty using a duality

principle and Monte Carlo method to estimate a lower bound for MI. Additionally MINE is differentiable because it is computed by neural networks.

Our proposed registration method uses this differentiable mutual information, MINE, so that the automatic differentiation of modern optimization toolboxes, such as PyTorch (Paszke et al., 2019) can be utilized. Additionally, our method computes transformation matrix via matrix exponential of a linear combination of basis matrices. We demonstrate experimentally that matrix exponential method yields more accurate registration. Our method also makes use of multi-resolution image pyramids. Unlike a conventional method where computation starts at the highest level of the image pyramid and gradually proceeds to the lower levels, we simultaneously use all the levels in gradient descent-based optimization using automatic differentiation. We refer to our proposed method as DRMIME (differentiable registration with mutual information and matrix exponential). DRMIME is able to achieve state-of-the-art accuracy on two benchmark data sets: FIRE (Hernandez-Matas et al., 2017) and ANHIR (ANH).

## 2. Background

### 2.1. Optimization for Image Registration

Let us denote by $T$ the fixed image and by $M$ the moving image to be registered. Let $H$ denote a transformation matrix signifying affine or any other suitable transformation. Further, let $Warp(M, H)$ denote a function that transforms the moving image $M$ by the transformation matrix $H$. Optimization-based image registration minimizes the following objective function to find the optimum transformation matrix $H$ that aligns the transformed moving image with the fixed image:

$$\min_{H} D(T, Warp(M, H)), \tag{1}$$

where $D$ is a loss function that typically measures a distance between the fixed and the warped moving image.

### 2.2. Matrix Exponential

The optimization problem (1) can be carried out by gradient descent when the loss function $D$ and $Warp$ are differentiable with respect to elements of $H$ that are not constrained. When the elements of matrix $H$ are constrained, as in the rigid-body transformation, matrix exponential provides a remedy for gradient descent. For example, finding the parameters for rigid transformation can be seen as an optimization problem on a finite dimensional Lie group (Schröter et al., 2010).

One of the earliest works (Taylor and Kriegman, 1994) shows how to perform optimization procedures over the Lie group $SO(3)$ and related manifolds. Later, Wachinger and Navab (Wachinger and Navab, 2013) showed the use of matrix exponential for image sequence registration. For brevity, here we just state the mapping for the $Aff(2)$ group, which is the group of affine transformations on the 2D plane. This group has 6 generators:

$$B_1 = \begin{pmatrix} 0 & 0 & 1 \\ 0 & 0 & 0 \\ 0 & 0 & 0 \end{pmatrix}, B_2 = \begin{pmatrix} 0 & 0 & 0 \\ 0 & 0 & 1 \\ 0 & 0 & 0 \end{pmatrix}, B_3 = \begin{pmatrix} 0 & 1 & 0 \\ 0 & 0 & 0 \\ 0 & 0 & 0 \end{pmatrix}, B_4 = \begin{pmatrix} 0 & 0 & 0 \\ 1 & 0 & 0 \\ 0 & 0 & 0 \end{pmatrix}, B_5 = \begin{pmatrix} 1 & 0 & 0 \\ 0 & -1 & 0 \\ 0 & 0 & 0 \end{pmatrix}, B_6 = \begin{pmatrix} 0 & 0 & 0 \\ 0 & -1 & 0 \\ 0 & 0 & 1 \end{pmatrix}.$$

If $v = [v_1, v_2, ..., v_6]$ is a parameter vector, then the affine transformation matrix is obtained using the expression: $Mexp(\sum_{i=1}^{6} v_i B_i)$, where $Mexp$ is the matrix exponentiation operation that can be computed as:

$$Mexp(A) = \sum_{n=0}^{\infty} \frac{A^n}{n!}, \tag{2}$$

for a matrix $A$. In DRMIME, we truncate the series after 10 terms and empirically find that this choice yields good registration accuracy. The image registration optimization defined in (1) now takes the following form:

$$\min_{v_1,...,v_6} D(T, Warp(M, Mexp(\sum_{i=1}^{6} v_i B_i))). \tag{3}$$

We can apply standard mechanisms of partial derivative $\frac{\partial D}{\partial v_i}$ computation by automatic differentiation (i.e., chain rule) and adjust parameter $v_i$ by gradient descent.

### 2.3. Multi-resolution Computation

Large displacements between the fixed and the moving images pose a significant challenge for the optimization that can be mitigated by the use of multi-resolution pyramids (Thevenaz et al., 1998; Krüger and Calway, 1998; Alhichri and Kamel, 2002), In multi-resolution method a pyramid of images is constructed where the original image lies at the bottom level and subsequent higher levels have down-scaled, Gaussian blurred versions of the image.

Using the multi-resolution recipe, two image pyramids are built: $T_l$ and $M_l$, $l = 1, ..., L$, where $L$ is the maximum level in the pyramid. $T_1 = T$ and $M_1 = M$ are the original fixed and moving images, respectively. The registration problem (3) takes the following form:

$$\min_{v_1,...,v_6} \sum_{l=1}^{L} D(T_l, Warp(M_l, Mexp(\sum_{i=1}^{6} v_i B_i))). \tag{4}$$

The usual practice for a multi-resolution approach is to start computation at the highest (i.e., coarsest) level of the pyramid and gradually proceed to the original resolution. In contrast, we found that working simultaneously on all the levels as captured in the optimization problem (4) is more beneficial.

However, note also that image structures are slightly shifted through multi-resolution image pyramids. So, a transformation matrix suitable for a coarse resolution may need a slight correction when used for a finer resolution. To mitigate this issue, we exploit matrix exponential parameterization and introduce an additional parameter vector $v^1 = [v_1^1, ..., v_6^1]$ exclusively for the finest resolution and modify optimization (4) as follows:

$$\min_{\substack{v_1,\cdots,v_6 \\ v_1^1,\cdots,v_6^1}} \{\sum_{l=2}^{L} D(T_l, Warp(M_l, Mexp(\sum_{i=1}^{6} v_i B_i))) + D(T_1, Warp(M_1, Mexp(\sum_{i=1}^{6}(v_i + v_i^1)B_i)))\}. \tag{5}$$

### 2.4. Mutual Information

Since different modalities can have different image intensities and varying contrast levels between them, it is unlikely that simply using Mean Square Error as a registration metric

will work well. This is why Mutual Information (MI) is commonly used in multi-modality registration. MI, in general, is defined as a measure of dependence between two random variables. In the context of image registration, this means that two initially unregistered images will have an MI score which is lower than the MI score between the images once they are completely registered. Gradient-based methods (Maes et al., 1997) for MI based image registration work quite well for such cases. In these implementations, MI between two random variables are computed by Kullback-Leibler (KL-) divergence (Kullback, 1997), which uses both joint and marginal probability densities.

For scalar-valued images, joint probabilities are calculated using a two-dimensional histogram of the two images. Most current MI-based techniques for registration use slight variations of the above method to approximate MI. While this works well, there are some issues associated with this method of evaluation as follows.

- The number of histogram bins chosen becomes a hyperparameter. While increasing the number of bins would lead to better accuracy in computation, this comes at the cost of time.

- Images with higher dimensions (e.g., color images), would need a higher dimensional histograms and a joint histogram requiring a very large sample size that is often computationally prohibitive. For instance, an RGB image with 3 channels would need a 6-dimensional joint histogram. A common way to bypass this restriction is to work with grayscale intensities of images, but this leads to loss of valuable information.

A potential solution to the above problems is presented by MINE (Belghazi et al., 2018) that uses the Donsker-Varadhan (DV) duality to compute MI (we provide a simple proof in the Appendix (Section 6.1)):

$$MI = \sup_f J(f), \tag{6}$$

where $J(f)$ is the DV lower bound:

$$J(f) = \int f(x,z)P_{XZ}(x,z)dxdz - log(\int exp(f(x,z))P_X(x)P_Z(z)dxdz), \tag{7}$$

where $P_{XZ}$ is the joint density for random variables $X$ and $Z$. $P_X$ and $P_Z$ are marginal densities for $X$ and $Z$, respectively. MINE uses a neural network to compute $f(x,z)$ and uses Monte Carlo technique to approximate (7). MINE claims that computations of (6) scales better than histogram-based computation of MI (Belghazi et al., 2018).

The optimization for image registration (5) using mutual information now becomes:

$$\max_{\substack{v_1,\cdots,v_6 \\ v_1^1,\cdots,v_6^1 \\ \theta}} \{\sum_{l=2}^{L} DV(T_l, Warp(M_l, Mexp(\sum_{i=1}^{6} v_i B_i))) + DV(T_1, Warp(M_1, Mexp(\sum_{i=1}^{6}(v_i+v_i^1)B_i)))\}, \tag{8}$$

where $\theta$ denotes the parameters of the neural network that MINE uses to realize $f$. Notation $DV(X,Z)$ in (8) is used to denote DV lower bound (7) computed on two images $X$ and $Z$.

## 3. DRMIME Algorithm

Fig. 1 shows a schematic for the optimization problem (8). Our proposed Algorithm 2 implements DRMIME that uses DV lower bound (7) computed in turn by Algorithm 1,

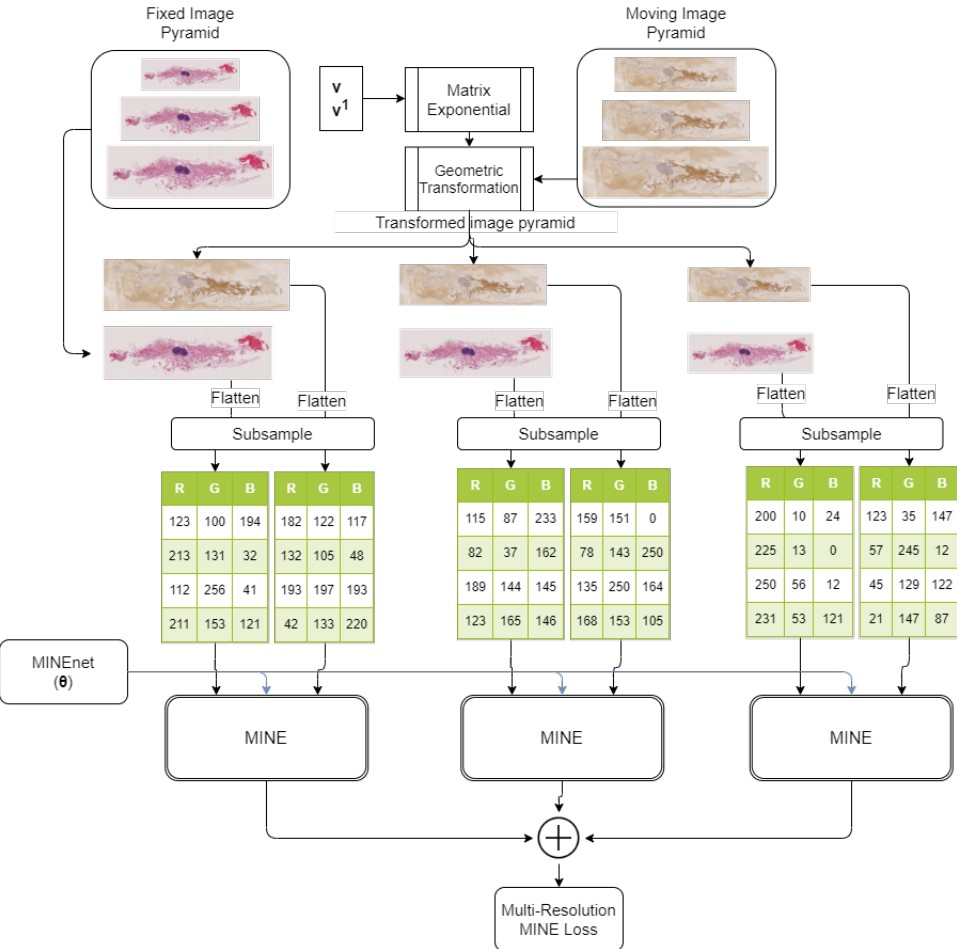

Figure 1: Pipeline for the DRMIME Registration algorithm

which employs a fully connected neural network $f_\theta$ MINEnet. MINEnet has two hidden layers with 100 neurons in each layer. We use ReLU non-linearity in both the hidden layers. Appendix contains details about implementation including learning rates, hyperparameters and optimizations used. The code for DRMIME is available here.

---

**Algorithm 1:** Mutual Information Neural Estimation

---

$MINE(X, Z, I)$

**Input:** Image $X$, Image $Z$, Sampled pixel locations $I$

**Output:** Estimated mutual information (DV lower bound)

    Shuffle pixel locations: $I^s = RandomPermute(I)$ ;

    $N = length(I)$ ;

    $DV = \frac{1}{N} \sum_j f_\theta(X_{I_j}, Z_{I_j}) - log(\frac{1}{N} \sum_j exp(f_\theta(X_{I_j}, Z_{I_j^s})))$ ;

    Return $DV$;

---

Algorithm 1 takes in two images $X$ and $Z$ along with a subset of pixel locations $I$. It creates a random permutation $I^s$ of the indices $I$. $I_j$ denotes the $j^{\text{th}}$ entry in the index list $I$, while $X_{I_j}$ denotes the $I_j^{\text{th}}$ pixel location on image $X$. Finally, the algorithm returns the DV lower bound (Belghazi et al., 2018) computed by Monte Carlo approximation of (7).

---

**Algorithm 2:** DRMIME

---

**Input:** Fixed image $T$, moving image $M$
**Output:** Transformation matrix $H_1$
Set learning rates $\alpha$, $\beta$, $\gamma$ and pyramid level $L$;
Build multiresolution image pyramids $\{T_l\}_{l=1}^L$ from $T$ and $\{M_l\}_{l=1}^L$ from $M$;
Use random initialization for MINEnet parameters $\theta$ ;
Initialize $v$ and $v^1$ to the 0 vectors ;
**for** *each iteration* **do**
    $MI = 0$ ;
    $H = Mexp(\sum_{i=1}^6 v_i B_i)$ ;
    $H_1 = Mexp(\sum_{i=1}^6 (v_i + v_i^1) B_i)$ ;
    $I_1 = $ Sample pixel locations on $T_1$ ;
    $MI \mathrel{+}= MINE(T_1, Warp(M_1, H_1), I_1)$ ;
    **for** $l = [2, L]$ **do**
        $I_l = $ Sample pixel locations on $T_l$ ;
        $MI \mathrel{+}= MINE(T_l, Warp(M_l, H), I_l)$ ;
    **end**
    Update MINEnet parameter: $\theta \mathrel{+}= \alpha \nabla_\theta MI$ ;
    Update matrix exponential parameters: $v \mathrel{+}= \beta \nabla_v MI$ and $v^1 \mathrel{+}= \gamma \nabla_{v^1} MI$;
**end**
Compute final transformation matrix: $H_1 = Mexp(\sum_{i=1}^6 (v_i + v_i^1) B_i)$ ;

---

Algorithm 2 builds two image pyramids, one for the fixed image $T$ and another for the moving image $M$. Due to memory constraints, especially for GPU, a few pixel locations are sampled that enter actual computations. This step appears as "Subsample" in Fig. 1. We have used two variations of sampling: (a) randomly choosing only 10% of pixels locations on each resolution and (b) finding Canny edges (Canny, 1986) on the fixed image and choosing only the edge pixels. Our ablation study shows a comparison between these two options. Fig. 1 illustrates two other computation modules- "Matrix Exponential" and "Geometric Transformation" that denote $Mexp$ and $Warp$ operations, respectively.

## 4. Datasets and Evaluation Metric

The datasets chosen for our experiments correspond to testing two important hypotheses. First, performing image registration with our algorithm on images within the same modality fares comparably (or better) to other standard algorithms. For this, we use the FIRE dataset (Hernandez-Matas et al., 2017). Second, since our algorithm is based on MI, it can handle multi-modal registration successfully as well. For this we use data from the

| Algorithm | NAED ($\mu \pm \sigma$) | p-value |
|---|---|---|
| DRMIME($v$) | **0.0048** $\pm$ 0.014 | - |
| DRMIME | **0.0048** $\pm$ 0.026 | - |
| NCC | 0.0194 $\pm$ 0.033 | 1.3e-04 |
| MMI | 0.0198 $\pm$ 0.034 | 5.4e-05 |
| NMI | 0.0228 $\pm$ 0.032 | 1.7e-08 |
| JHMI | 0.0311 $\pm$ 0.046 | 4.5e-07 |
| AMI | 0.0441 $\pm$ 0.028 | 1.4e-27 |
| MSE | 0.0641 $\pm$ 0.094 | 3.5e-03 |

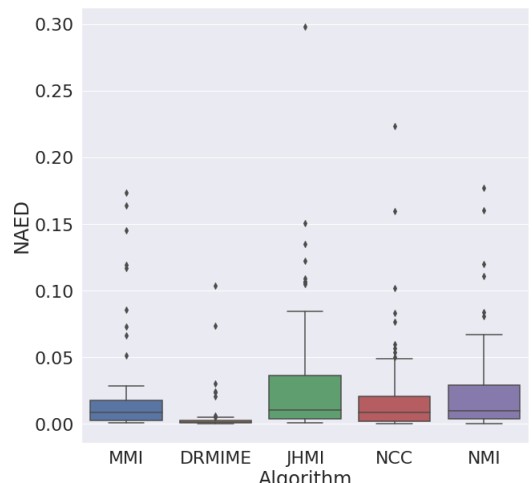

Table 1: NAED for FIRE dataset along with paired t-test significance values

Figure 2: Box plot for NAED of the best 5 performing algorithms on FIRE

ANHIR (Automatic Non-rigid Histological Image Registration) 2019 challenge (ANH). Both datasets contain color images.

The FIRE dataset (Hernandez-Matas et al., 2017) provides 134 retinal fundus image pairs divided into 3 categories: S, P and A. The primary uses of the categories being Super Resolution, Mosaicing and Longitudinal Study, respectively. The dataset states that while categories S and A have $> 75\%$ overlap, category P has very little overlap ($< 75\%$); so none of the algorithms we evaluated (including ours) perform well on P category, leading to little or no registration in most cases (even diverging in some instances). So for a fair evaluation, we leave out category P.

The ANHIR dataset (ANH) provides pairs of 2D microscopy images of histopathology tissue samples stained with different dyes. The task is difficult due to non-linear deformations affecting the tissue samples, different appearance of each stain, repetitive texture, and the large size of the whole slide images.

The FIRE dataset provides the location of 10 points in each image and the location of the corresponding 10 points in the paired (to-be-registered) image. These points were obtained by annotations created by experts and further refined to mitigate human error. ANHIR dataset usually contains more than 10 ground truth points.

Once we obtain the transformation matrix, we transform ground truth points on the moving image and compute the Euclidean distance between these transformed points and the ground truth points on the fixed image. Further we normalize these distances between 0 and 1 and call this metric Normalized Average Euclidean Distance (NAED). In our evaluation we used NAED for both the datasets. For ANHIR, only 230 pairs are available with their ground truth as part of the training data, so we only evaluate on this set of images.

| Algorithm | NAED ($\mu \pm \sigma$) | p-value |
|---|---|---|
| DRMIME($v$) | $0.0393 \pm 0.081$ | - |
| DRMIME | $\mathbf{0.0384} \pm 0.087$ | - |
| NCC | $0.0461 \pm 0.084$ | 7.0e-04 |
| MMI | $0.0490 \pm 0.082$ | 6.2e-05 |
| MSE | $0.0641 \pm 0.094$ | 5.5e-14 |
| NMI | $0.0765 \pm 0.090$ | 3.0e-31 |
| AMI | $0.0769 \pm 0.090$ | 3.7e-30 |
| JHMI | $0.0827 \pm 0.100$ | 8.3e-21 |

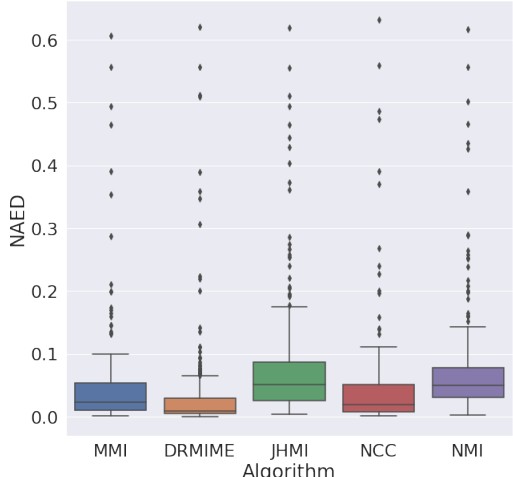

Table 2: NAED for ANHIR dataset along with paired t-test significance values

Figure 3: Box plot for NAED of the best 5 performing algorithms on ANHIR

## 5. Experiments and Discussion

Competing algorithms were selected based on whether they use MI or can be used for multi-modal registration. More information about these algorithms can be found in the Appendix 6.2: (a) Mattes Mutual Information **(MMI)**, (b) Joint Histogram Mutual Information **(JHMI)**, (c) Normalized Cross Correlation **(NCC)**, (d) Mean Square Error **(MSE)**, (e) AirLab Mutual Information **(AMI)**, (f) Normalized Mutual Information **(NMI)**.

While it is possible to use perspective transforms with DRMIME by changing the coefficient vector dimension and generators for matrix exponential, in order to have a fair comparison, we limit our algorithm to affine transform, because most libraries do so. The implementations of the above algorithms were used from these packages: **SITK**: MMI, JHMI, NCC, MSE. **AirLab**: AMI. **SimpleElastix**: NMI

For all evaluations, we also conduct a paired t-test with DRMIME to investigate if the results are statistically significant (p-value $< 0.05$). Fig. 4 and Fig. 5 in the Appendix show registration results for two random samples from FIRE and ANHIR datasets, respectively.

### 5.1. Accuracy

Table 1 shows the NAED for all algorithms on the FIRE dataset. Here, DRMIME performs almost an order of magnitude better than the competing algorithms and the results are statistically significant. Fig. 2 presents box plots the same metrics from Table 1. We note that DRMIME has very few outliers due to the robustness of the algorithm. Table 1 also shows a variation DRMINE($v$) that does not use the finetuning coefficients $v^1$ for the finest resolution. For FIRE dataset, we do not notice any difference these two versions.

Table 2 presents the NAED metrics for the ANHIR dataset. While the margin of improvement is not as large as in case of the FIRE dataset, DRMIME is still statistically the best performing algorithm. The box-plots in Fig. 3 also emphasise the same conclusion

as we saw before, i.e. DRMIME outperforms the other competing algorithms. DRMIME algorithm using $v^1$ shows a slight improvement in accuracy for the ANHIR dataset.

## 5.2. Efficiency

On a set of 10 randomly selected images (the set remains the same across all algorithms) from the FIRE dataset, we run these two sets of experiments for all the algorithms. We report the registration accuracy in terms of the ground truth (NAED) of these 10 images. The hardware for these experiments was NVIDIA GeForce GTX 1080 Ti, Intel(R) Xeon(R) CPU E5-2620 v4 @ 2.10GHz, 32GB RAM.

We run each algorithm for 1000 epochs, report the time taken and the accuracy achieved. The time taken tells us the fastest algorithm among those being considered, and also at the same time, its accuracy should at least be on par with other slower algorithms.

| Algorithm | Time (seconds) | NAED |
|---|---|---|
| DRMIME (50 epochs) | **58** | 0.02037 |
| NMI | 60 | 0.02503 |
| AMI | 620 | 0.02942 |
| DRMIME | 1425 | **0.00368** |
| MMI | 2904 | 0.00598 |
| JHMI | 1859 | 0.00605 |
| NCC | 3804 | 0.00697 |
| MSE | 2847 | 0.02918 |

Table 3: Time taken for 1000 epochs and resultant NAED (lower is better)

From Table 3, we can infer that while our algorithm attains the best NAED, it ranks third in terms of time taken to execute 1000 epochs. While AMI and NMI are faster, they are almost an order of magnitude worse in terms of the NAED performance.

Since this is a tradeoff between time and efficiency, DRMIME can perform extremely well at both ends of the spectrum. For instance, while individual epochs on AMI and NMI might be faster, we can achieve comparable accuracy by running DRMIME for much less epochs; within 50 epochs of optimization DRMIME achieves an NAED of 0.02037 taking only 58 seconds. The reason for a single epoch taking longer for DRMIME can be attributed to the fact that it works with batched data.

Also as a note, only DRMIME and AMI are GPU compatible, while the remaining algorithms run on CPU.

## 5.3. Ablation study

In this section we perform several ablation studies to have an understanding of the roles of all the components used in DRMIME, such as multi-resolution pyramids, matrix exponential and smart feature selection via Canny edge detection. We compare the performance of DRMIME to versions of it without using the aforementioned components.

## 5.3.1. Effect of multi-resolution

All hyperparameters are kept the same in the with and without experiments, the only difference being in the with multi-resolution experiment we use 6 levels of the Gaussian pyramids in the DRMIME algorithm, whereas in the without experiment we have a single level which is the native resolution of the image. Table 4 lists the results for these experiments.

| Dataset | DRMIME | Without MultiRes | p-value |
|---------|--------|------------------|---------|
| FIRE | $0.0048 \pm 0.014$ | **0.0043** $\pm 0.014$ | 0.365 |
| ANHIR | **0.0384** $\pm 0.087$ | $0.1089 \pm 0.150$ | 1.78e-15 |

Table 4: NAED for MINE with and without using multi-resolution pyramids

While the idea of multi-resolution was introduced in image registration to facilitate optimization, we note that many of the off-the-shelf algorithms have the same learning rate for all levels. As we are working with only an approximation of the distribution of the original data at different levels of the pyramid, there is a small chance that optimization at a particular sublevel could diverge. This leads to poor registration results occasionally. In our implementation of DRMIME, we produce batches which include data from all levels of the pyramid, making the optimization process much more robust, faster and less prone to divergence. Fig. 2 provides evidence to this since very few results fall outside the interquartile range (as compared to other algorithms).

## 5.3.2. Effect of matrix exponentiation

All hyperparameters are again kept the same in the with and without experiments; the only difference being, that rather than using a manifold basis vector, we now have 6 parameters indicating the degrees of freedom of an affine transform in a transformation matrix, i.e.

$$\begin{pmatrix} \theta_1 & \theta_2 & \theta_3 \\ \theta_4 & \theta_5 & \theta_6 \\ 0 & 0 & 1 \end{pmatrix}.$$

| Dataset | DRMIME | Without Manifolds | p-value |
|---------|--------|-------------------|---------|
| FIRE | $0.0048 \pm 0.014$ | **0.0045** $\pm 0.015$ | 0.4933 |
| ANHIR | **0.0384** $\pm 0.087$ | $0.0580 \pm 0.134$ | 0.0012 |

Table 5: NAED for MINE with and without using matrix exponentiation

Table 5 presents the results for these experiments. While the ablation study on the FIRE dataset results in similar results, the p-values from the paired t-test tells us that the results are not very significant to be able to interpret anything. The ANHIR dataset on the other hand sees a statistically significant improvement with use of matrix exponentiation.

### 5.3.3. Effect of Sampling strategy

It could be argued that our smart feature extraction via Canny edge detection helps DR-MIME perform better than other algorithms, since other algorithms do not have such custom feature detectors embedded in their pipeline. In order to reduce this potential confounding variable, we also assessed the performance of DRMIME with random sampling as well to make a fair comparison.

| Dataset | With Canny | Random Sampling(10%) | p-value |
|---------|------------|----------------------|---------|
| FIRE | **0.0048** $\pm$ 0.014 | 0.0097 $\pm$ 0.026 | 0.0296 |
| ANHIR | **0.0384** $\pm$ 0.087 | 0.0588 $\pm$ 0.167 | 0.0333 |

Table 6: NAED for MINE with Canny edge detection and Random Sampling (10%)

Table 6 presents these results. As we can be seen, there is a small drop in performance, but DRMIME still performs better than all the other algorithms with FIRE (Table 1) and better than most other algorithms with ANHIR (Table 2). This comes at a small cost of the optimizer taking longer to converge. It is important to note, that DRMIME results are using only 10% sampling due to limited memory available on the GPU, whereas the other algorithms use 50% sampling (see Appendix).

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

ITK. itk::jointhistogrammutualinformationimagetoimagemetricv4< tfixedimage, tmovingimage, tvirtualimage, tinternalcomputationvaluetype, tmetrictraits > class template reference, a. URL https://itk.org/Doxygen/html/classitk{_}1{_}1JointHistogramMutualInformationImageToImageMetricv4.html.

ITK. itk::mattesmutualinformationimagetoimagemetricv4< tfixedimage, tmovingimage, tvirtualimage, tinternalcomputationvaluetype, tmetrictraits > class template reference, b. URL https://itk.org/Doxygen/html/classitk{_}1{_}1MattesMutualInformationImageToImageMetricv4.html.

ITK. itk::meansquaresimagetoimagemetricv4< tfixedimage, tmovingimage, tvirtualimage, tinternalcomputationvaluetype, tmetrictraits > class template reference, c. URL https://itk.org/Doxygen/html/classitk{_}1{_}1MeanSquaresImageToImageMetricv4.html.

ITK. itk::normalizedcorrelationimagetoimagemetric< tfixedimage, tmovingimage > class template reference, d. URL https://itk.org/Doxygen/html/classitk{_}1{_}1NormalizedCorrelationImageToImageMetric.html.

Stefan Klein and Marius Staring. itk::advancedimagetoimagemetric< tfixedimage, tmovingimage > class template reference. URL http://elastix.isi.uu.nl/doxygen/classitk{_}1{_}1AdvancedImageToImageMetric.html.

Stefan Krüger and Andrew Calway. Image registration using multiresolution frequency domain correlation. In *BMVC*, pages 1–10, 1998.

Solomon Kullback. *Information theory and statistics*. Courier Corporation, 1997.

Frederik Maes, Andre Collignon, Dirk Vandermeulen, Guy Marchal, and Paul Suetens. Multimodality image registration by maximization of mutual information. *IEEE transactions on Medical Imaging*, 16(2):187–198, 1997.

David Mattes, David R Haynor, Hubert Vesselle, Thomas K Lewellyn, and William Eubank. Nonrigid multimodality image registration. In *Medical Imaging 2001: Image Processing*, volume 4322, pages 1609–1620. International Society for Optics and Photonics, 2001.

David Mattes, David R Haynor, Hubert Vesselle, Thomas K Lewellen, and William Eubank. Pet-ct image registration in the chest using free-form deformations. *IEEE transactions on medical imaging*, 22(1):120–128, 2003.

Adam Paszke, Sam Gross, Francisco Massa, Adam Lerer, James Bradbury, Gregory Chanan, Trevor Killeen, Zeming Lin, Natalia Gimelshein, Luca Antiga, Alban Desmaison, Andreas Kopf, Edward Yang, Zachary DeVito, Martin Raison, Alykhan Tejani, Sasank Chilamkurthy, Benoit Steiner, Lu Fang, Junjie Bai, and Soumith Chintala. Pytorch: An imperative style, high-performance deep learning library. In H. Wallach, H. Larochelle, A. Beygelzimer, F. d'Alché-Buc, E. Fox, and R. Garnett, editors, *Advances in Neural Information Processing Systems 32*, pages 8024–8035. Curran Associates, Inc., 2019. URL http://papers.neurips.cc/paper/9015-pytorch-an-imperative-style-high-performance-deep-learning-library.pdf.

J. P. W. Pluim, J. B. A. Maintz, and M. A. Viergever. Mutual-information-based registration of medical images: a survey. *IEEE Transactions on Medical Imaging*, 22(8): 986–1004, Aug 2003. ISSN 1558-254X. doi: 10.1109/TMI.2003.815867.

Robin Sandkühler, Christoph Jud, Simon Andermatt, and Philippe C. Cattin. Airlab: Autograd image registration laboratory. *CoRR*, abs/1806.09907, 2018. URL http://arxiv.org/abs/1806.09907.

Martin Schröter, Uwe Helmke, and Otto Sauer. A lie-group approach to rigid image registration. *arXiv preprint arXiv:1007.5160*, 2010.

Colin Studholme, Derek LG Hill, and David J Hawkes. An overlap invariant entropy measure of 3d medical image alignment. *Pattern recognition*, 32(1):71–86, 1999.

Camillo J Taylor and David J Kriegman. Minimization on the lie group so (3) and related manifolds. *Yale University*, 16(155):6, 1994.

Philippe Thévenaz and Michael Unser. Optimization of mutual information for multiresolution image registration. *IEEE transactions on image processing*, 9(ARTICLE):2083–2099, 2000.

Philippe Thevenaz, Urs E Ruttimann, and Michael Unser. A pyramid approach to subpixel registration based on intensity. *IEEE transactions on image processing*, 7(1):27–41, 1998.

Paul Viola and William M Wells III. Alignment by maximization of mutual information. *International journal of computer vision*, 24(2):137–154, 1997.

C. Wachinger and N. Navab. Simultaneous registration of multiple images: Similarity metrics and efficient optimization. *IEEE Transactions on Pattern Analysis and Machine Intelligence*, 35(5):1221–1233, 2013.

## 6. Appendix

### 6.1. DV Lower Bound Reaches Mutual Information

MINE maximizes the DV lower bound (7) with respect to a function $f(x, z)$. Let us consider a perturbation function $g(x, z)$ and the perturbed objective function $J(f + \epsilon g)$ for a small number $\epsilon$. Taking the following limit (using L'Hospital's rule), we obtain:

$$\lim_{\epsilon \to 0} \frac{J(f + \epsilon g) - J(f)}{\epsilon} = \int g(x, z)[P_{XZ}(x, z) - \frac{exp(f(x, z))P_X(x)P_Z(z)}{\int exp(f(x, z))P_X(x)P_Z(z)dxdz}]dxdz. \quad (9)$$

Using principles of calculus of variations(Gelfand et al., 2000), this limit should be 0 for $J$ to achieve an extremum. Since perturbation function $g(x, z)$ is arbitrary, this condition is possible only when

$$P_{XZ}(x, z) = \frac{exp(f(x, z))P_X(x)P_Z(z)}{\int exp(f(x, z))P_X(x)P_Z(z)dxdz}, \quad (10)$$

i.e., the Gibbs density (Belghazi et al., 2018) is achieved. From (10), we obtain:

$$f(x, z) = log(\frac{P_{XZ}(x, z)}{P_X(x)P_Z(z)} \int exp(f(x, z))P_X(x)P_Z(z)dxdz). \quad (11)$$

Using this expression in equation (7), we obtain:

$$J(f) = \int P_{XZ}(x, z)log\frac{P_{XZ}(x, z)}{P_X(x)P_Z(z)}dxdz. \quad (12)$$

Thus, maximization of $J(f)$ leads to mutual information.

### 6.2. Competing algorithms

1. **Mattes Mutual Information (MMI)** (Mattes et al., 2001, 2003; ITK, b): MI is usually defined as:

   $$MI = \int p_{XY}(x, y) \log \frac{p_{XY}(x, y)}{p_X(x)p_Y(y)}dxdy, \quad (13)$$

   According to equation (13), we need to compute the joint ($p_{XY}$) and marginal ($p_X, p_Y$) probabilities of the fixed and moving images. To reduce the effects of quantization from interpolation and discretization due to binning, this version of MI computation uses Parzen windowing to form continuous estimates of the underlying image histogram.

2. **Joint Histogram Mutual Information (JHMI)** (Thévenaz and Unser, 2000; ITK, a): This method computes Mutual Information using Parzen windows as well, but it uses separable Parzen windows. By selection of a Parzen window that satisfies the partition of unity, it provides a tractable closed-form expression of the gradient of the MI computation with respect to transformation parameters.

3. **Normalized Cross Correlation (NCC)**(ITK, d): As the names says, the correlation between the moving and the fixed image pixel intensities is computed. The correlation is normalized by the autocorrelations of both the fixed and moving images.

4. **Mean Square Error (MSE)**(ITK, c): This is the mean squared difference of the pixelwise intensity between the fixed and moving image.

5. **AirLab Mutual Information (AMI)**(Sandkühler et al., 2018): AirLab is a PyTorch based image registration framework. It performs histogram based mutual information computation(Viola and Wells III, 1997; Maes et al., 1997). Since it is a deep learning based solution, it provides support for using batches as well as state-of-the-art optimizers and GPU support.

6. **Normalized Mutual Information (NMI)**(Studholme et al., 1999; Klein and Staring): The initial PDF (probability density function) construction is done using Parzen histograms, and then MI is obtained by double summing over the discrete PDF values. In this metric, the final MI is normalized to a range between 0 and 1.

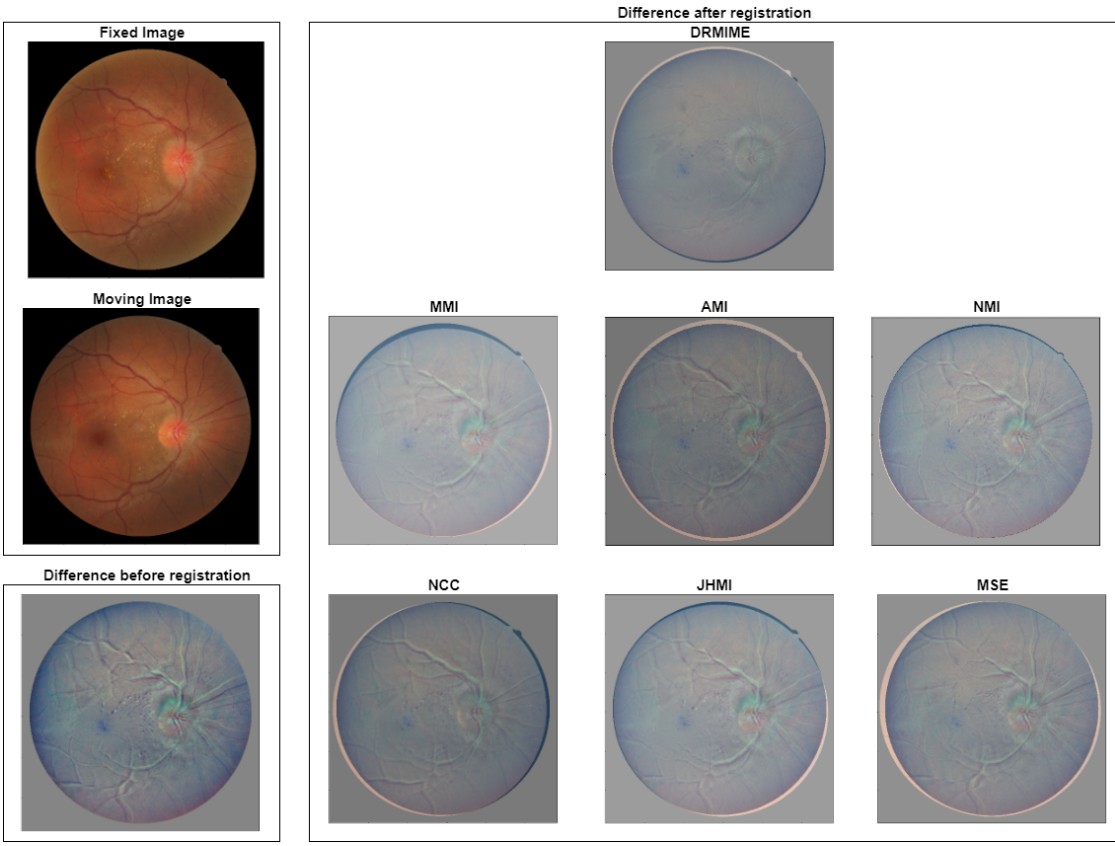

Figure 4: The images on the left show a pair to be registered from the FIRE dataset. The images on the right represent the difference between the transformed moving image and the fixed image after registration by different algorithms.

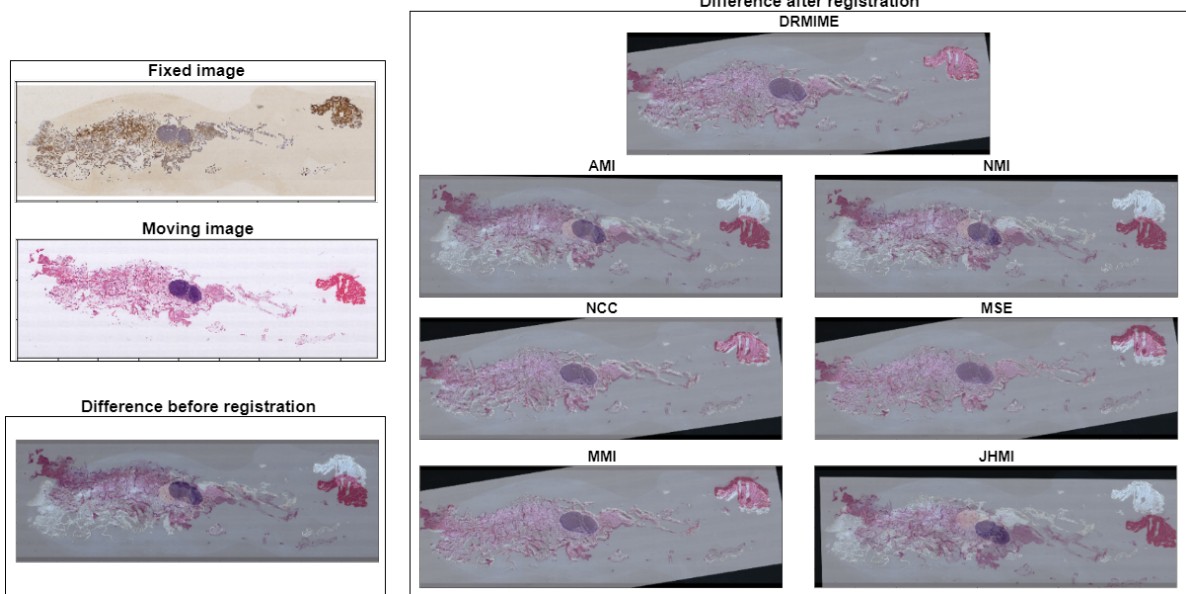

Figure 5: The images on the left show a pair to be registered from the ANHIR dataset. The images on the right represent the difference between the transformed moving image and the fixed image after registration by different algorithms.

## 6.3. Preprocessing

### 6.3.1. FIRE

Each image in this dataset is $2912 \times 2912$ pixels, but only the central portion of the images contain the retinal fundus, the rest of the image being black. While it's possible to use masks to remedy this, not all frameworks support masks, so in order to have a fair comparison across all algorithms, we crop these images to include only the retinal fundus. The cropping was selected such that it includes no blank (black) space and it remains rectangular (square). The cropped area was $1941 \times 1941$ pixels.

### 6.3.2. ANHIR

The ANHIR dataset has extremely high resolution pictures (some categories go upto $65k \times 60k$ pixels on average) and some registration frameworks fail to process such large images. Furthermore, different stainings of the same tissue have different resolutions as well. To solve these two problems when registering a pair of images, they are scaled down by a factor of 5 while keeping the original aspect ratio; this solves the first problem. Then the image with the smaller aspect ratio is rescaled to match the width of the image with the larger aspect ratio and the top and bottom of the smaller one are padded to match the height of the larger. This way we keep the aspect ratio of the original images with no distortions and still arrive at a common and smaller, more manageable resolution.

### 6.4. Hyperparameters

All architectures and hyper-parameters for our experiments are listed here:

**DRMIME**:

- learningRate: $\alpha = 1e - 3$, $\beta = 5e - 3$, $\gamma = 1e - 4$
- number of pyramid levels $L = 6$
- numberOfIterations: 500 (FIRE)/1500 (ANHIR)
- Optimizer : ADAM with AMSGRAD

**MMI**:

1. learningRate: 1e-5
2. numberOfIterations: 5000
3. numberOfHistogramBins: 100
4. convergenceMinimumValue: 1e-9
5. convergenceWindowSize: 200
6. SamplingStrategy: Random
7. SamplingPercentage: 0.5

**JHMI**:

1. learningRate: 1e-1
2. numberOfIterations: 5000
3. numberOfHistogramBins: 100
4. convergenceMinimumValue: 1e-9
5. convergenceWindowSize: 200
6. SamplingStrategy: Random
7. SamplingPercentage: 0.5

**MSE**:

1. learningRate: 1e-6
2. numberOfIterations: 5000
3. convergenceMinimumValue: 1e-9
4. convergenceWindowSize: 200

**NCC**:

1. learningRate: 1e-1
2. numberOfIterations: 5000
3. convergenceMinimumValue: 1e-9
4. convergenceWindowSize: 200

**NMI**:

1. numberOfIterations: 5000

**AMI**:

1. learningRate: 1e-4
2. numberOfIterations: 5000
3. bins: 64
4. sigma: 3
5. spatialSamples: 0.1
6. Optimizer : AMSGRAD

