# OpenReview forum: "DRMIME: Differentiable Mutual Information and Matrix Exponential for Multi-Resolution Image Registration"
_MIDL.io/2020/Conference — MIDL 2020_

### Official Review · AnonReviewer1 · 2020-03-05
**This paper presents a new and innovative MI-based registration method that can be applied to both multi- and mono-modal images.**

**Rating:** 3
**Confidence:** 5
**Recommendation:** Poster

**Summary:**

This paper presents a new registration method based on differentiable mutual information (MI). The first novelty of this paper is to use a recently proposed MI estimation method called MINE (mutual information neural estimation) that can estimate a lower bound of MI. Importantly, MINE is differentiable and so this can circumvent the drawbacks of traditional histogram-based MI computation. The second novelty of this work is to use the transformation matrix via matrix exponential of a linear combination of basis matrices. The experimental results demonstrate superior registration performance over other traditional methods.

**Strengths:**

-The use of the differentiable mutual information tackles some of the drawbacks of traditional histogram-based MI computation.
-The proposed method computes the transformation matrix via matrix exponential of a linear combination of basis matrices, which accounts for affine transformation.
-Matrix exponential makes the optimization process smoother.
-The proposed method was evaluated on currently accepted standard similarity measures, demonstrating superior performance.


**Weaknesses:**

In the method, the transformation is designed to capture affine transformation only, and extending it to deformable one may not be straightforward. In the validation part, it is necessary to show the advantages of both MINE part and the transformation
-

**Justification Of Rating:**

This work has sufficient contributions (e.g., the computation of mutual information alongside the design of the transformation matrix to account for the affine transformation) to the registration community to move forward.

**Paper Type:**

both

**Special Issue:**

no

---

> ### Author Response · Authors · 2020-03-27
> **Response**
>
> We have started working using MINE for deformable image registration. Results will be reported in a different venue.
>
> The question is not clear. We assuming that the revier is wondering about the results when matrix exponential is used with standard loss function such as MSE. We have done that experiment on FIRE dataset and found that MINE still yields better results. We have note reported it here. This can be added as an ablation study.

---

### Official Review · AnonReviewer2 · 2020-03-09
**Multimodal rigid registration using a N.N. based approximation of the Mutual Information**

**Rating:** 3
**Confidence:** 4

**Summary:**

In this paper, the authors present a new multimodal image registration algorithm which measures the similarity of the registered images using and approximation of the mutual information. The contribution of the paper is to use the neural networks-based mutual information approximation method of (Belghazi et al., 2018) for this approximation. They also make it multi-scale by comparing the images at several scales simultaneously.

The presented similarity metric is shown to compare well with other classic metrics such as the Mattes Mutual Information, the Normalized Mutual Information or the Normalized Cross Correlation for instance, on the FIRE dataset (Hernandez-Matas et al., 2017).

**Strengths:**


I think that the idea to learn on the fly an optimal similarity metric to compare multi-modal images at different scales simultaneously is interesting.

The results also seem promising.

The bibliography is sufficient for a conference paper.

**Weaknesses:**


-> Compared with Eq. (4), the effect of the additional term in Eq. (5) is far to be clear. Its effect should be assessed at least in appendix. Why also using the additional term $v_i^1$ at the first scale only? How to constraint them to remain small?

-> Although the parameters of Eqs. (4) and (5) are clear, the parameters $\theta$ of Eq (8) optimized by the N.N. are not clear. Are they the same for all scales? If yes, how to handle simultaneously images at different resolutions with the same parameters? What is also the meaning of this similarity metric which compares different representations of the same image.

-> What is the meaning of $f$ in this paper? To be honest, I can't understand from the paper what is optimized by the neural network in practice. Clear discussions should be given to explain the link between the formalism of MINE and the  optimized energy Eq. (8).

-> In the results section, the results obtained Table 1 using the approximation of the mutual information are about four time more accurate than those obtained using the true mutual information. This is really suspicious from a scientific point of view. It deserves extensive discussions.



**Justification Of Rating:**


The paper addresses an interesting question and the result look promising (though suspicious). I think the the authors should clarify their methodology before presenting it at MIDL. The meaning of the learned multiscale similarity metric is indeed not clear at all.

**Paper Type:**

both

**Questions To Address In The Rebuttal:**


Please address the issues raised above.

**Special Issue:**

no

---

> ### Author Response · Authors · 2020-03-27
> **Response**
>
> We can add an ablation study in the final version in the appendix that compares having V^1 vs. not having it. The learning rate for v^1 is an order of magnitude less than that for v. We explained the rationale to add this extra variable in the paragraph that precedes equation (5).
>
> We agree that this description should be made clear. f is the same as MINEnet_{\theta} in Algorithm 1.
>
> Through this work, we have realized that MINE is a very powerful tool for mutual information computation using duality principle. However, this should not be completely surprising, because similar duality principle is used in Wasserstein GANs that proves to be every useful. Since the reviewer expressed skepticism about results, we invite the reviewer to reproduce the results because our codes are released (https://github.com/abnan/DRMIME).
>
> All other methods use some sort of proxy for MI estimation since true MI being histogram based is not differentiable. For example, most other methods use Parzen windowing to estimate MI.
>
> We have started working using MINE for deformable image registration. Results will be reported in a different venue.
>
> The question is not clear. We are assuming that the reviewer is wondering about the results when matrix exponential is used with standard loss function such as MSE. We have done that experiment on FIRE dataset and found that MINE still yields better result.

---

### Official Review · AnonReviewer3 · 2020-03-09
**DRMIME review**

**Rating:** 2
**Confidence:** 4

**Summary:**

The paper address the image registration problem by using the existing MINE neural estimator for MI and matrix exponential for transoformation matirix. The novelty is to utilize the existing MINE neural estimator for mutual information computation and matrix exponential for rigid body transformation optimization for image registration application. The experiments are shown on ANH and FIRE datasets with standard NAED metric.

**Strengths:**

- The paper is very easy to follow.
- The idea of using the MINE neural estimator for MI in case of multi-modal image registration is straightforward. The idea utilizes the end to end neural network for image registration problem.
- Shows a way of overcoming histogram based MI estimation method by using MINE approximation method using neural network for image registration applications.


**Weaknesses:**

While the paper focused on MI metric for image registration, the paper lacks comparisons and citations to a wide range of recent deep learning based image registration algorithms in literature. For e.g.
- "Semi-Supervised Deep Metrics for Image Registration" Alireza et al 2018.
- "Deep similarity learning for multimodal medical images" Cheng et al 2016.
- "Networks for Joint Affine and Non-parametric Image Registration", Shen et al 2019,
- "Recursive Cascaded Networks for Unsupervised Medical Image Registration" Zhao et al 2019

The above mentioned papers for me are quite relevent to the motivation of the paper and show the comparisons of different metrics utilized in image registration for other than MI. The experiments were done only to basic metrics. Also the number of bins in AMI (default 64) and sigma is not mentioned in the comparison. These two parameters are quite sensitive and may change the reported results which needs to be fine tuned for the reported result.
Also  sentence like "MINE is differentiable because it is computed by neural networks" is very unwieldy for a scientific paper.




**Justification Of Rating:**

1. The paper is not much novel but utlizes the existing Mine algorithm for a MI metric based image registration
2. The paper lacks proper validation and comparison with respect to recent neural network based image registration methods.
3. The paper lacks citations to existing vast literature on similarity based image registration methods.

**Paper Type:**

validation/application paper

**Questions To Address In The Rebuttal:**

Cite and compare papers mentioned above.
What is the number of levels (L) used in pyramids built and discusse whether it is crucial w.r.t. the compared methods.
What is the average time by DRMIME for a complete image registration.

**Special Issue:**

no

---

> ### Author Response · Authors · 2020-03-27
> **Response**
>
> We chose to limit our comparisons only with optimization-based methods, while the deep learning-based methods mentioned by the reviewer are all learning-based requiring significant amount of training data.
>
> We will mention all the hyperparameter settings for AMI in the Appendix. Note that we have reported many other hyper-parameters in the Appendix. The sensitivity of these parameters in AMI for MI computation is yet another reason why MINE should be used instead. Other than the learning rate and the neural network architecture, which is a fully connected small shallow net, no hyper parameters are there in our method.
>
> We provide the time for 1000 epochs in Table 3. And for each algorithm, we state the number of epochs used (in Appendix). Total time can be inferred easily.
>
> We have used L = 6. We will mention this in the Appendix. Thanks for pointing out.

---

> > ### Comment · AnonReviewer3 · 2020-04-04
> > **Discussion-1**
> >
> > Change initial score into: Weak accept.

---

### Official Review · AnonReviewer4 · 2020-03-11
**A novel registration algorithm is proposed that combines three methodological ideas.**

**Rating:** 2
**Confidence:** 5

**Summary:**

A novel registration algorithm is proposed. The main innovative element is the use of a recently proposed method to estimate mutual information (Belghazi et al). It's a very good idea to test this approach in the context of medical image registration. Two other novel elements of the method that are highlighted in the paper are the transformation model (using matrix exponential to parameterize a rigid transformation matrix), and a particular multiresolution approach. The method is compared to 6 different algorithms on two public datasets with ground truth. The performance is promising.

**Strengths:**

- It is an interesting idea to apply the method of Belghazi to image registration.
- The transformation model based on matrix exponential seems elegant.
- The method is compared with 6 other algorithms.
- The method is evaluated on two public data sets (FIRE and ANHIR).

**Weaknesses:**

- The results of the comparison with other methods are difficult to interpret because it's not only the similarity measure that's different across methods, but also many other algorithmic components, such as optimisation method, transformation model, multiresolution approach, number of image samples to compute the similarity measure, etc.
- The manuscript lacks focus because it introduces three contributions at once. The ablation test in the appendix sheds some light on their individual added value, but such information is essential and should not have been hidden in the appendix.
- Eq 5, which assigns a separate transformation to the first level of the multiresolution pyramid seems quite ad hoc: why only for the first level? And if there is a separate transformation for the first level, then how does it still influence the other levels? It seems that the estimation of v^1 becomes independent from the estimation of v.
- The paper lacks some references and related discussion. Matrix exponential for modelling transformation was already proposed by Wachinger & Navab, IEEE PAMI 2013. Simultaneous multiresolution for image registration was investigated by Sun et al, IEEE Transactions on Image Processing, 2013.
- The paper is essentially too long, and the appendix contains information that should have been part of the main manuscript. Timing results and the ablation study are examples.
- The comparison of timing results is confusing because of the many differences in implementation details between methods.

**Detailed Comments:**

It's not clear whether the transformation model describes a rigid (rotation + translation) transformation, or an affine (rotation, translation, shear, scale). The text needs to be clarified on this aspect.

**Justification Of Rating:**

The idea to test this novel algorithm for estimating mutual information in an image registration framework is good, but the current manuscript raises too many concerns regarding experiment design and clarity of presentation.

**Paper Type:**

methodological development

**Questions To Address In The Rebuttal:**

I do not think that my concerns can be addressed by a rebuttal. My main concern is that the experiments should have compared the different implementations of mutual information within a single software framework, to allow fair comparison.

**Special Issue:**

no

---

> ### Author Response · Authors · 2020-03-27
> **Response**
>
> The intention of our work was to create an optimization-based method that yields state-of-the-art (SOTA) accuracy for image registration using rigid-body/affine/perspective transform and mutual information. That is why we compared with six leading off-the-shelf toolboxes that uses optimization, mutual information and these transforms.
>
> To test our hypothesis, we experimented with two publicly available datasets for which landmark-based ground truth is available. The intention of the study was not to examine every component we used in our algorithm. Many of these comparisons already exist in the literature. For example, MINE vs. other methods of mutual information computation exists in Belghazi et al.'s work. Advantage of using matrix exponential is discussed in Wachinger & Navab's work. Thus, our work has collected some of the best practices and used latest optimization tools (automatic gradients) to create a SOTA toolbox. Missing literature as pointed out in the reviews will be incorporated in the final version. However, we did offer a number of ablation studies that shone some light on our design choices.
>
> Multiresolution approach for other frameworks was tried out as a hyperparameter and almost in all cases the best performance was observed in case of using no multiresolution pyramid, it was just a factor which reduced time for the optimizer to converge. On the other hand our combined multiresolution approach improved robustness. Sampling comparision in our methodology was switched to random sampling too in the ablation study.
>
> The paragraph preceding equation (5), explains the use of two parameter vectors v and v^1. We use v for all the resolution levels, while v^1 is used only for the finest resolution. This can be viewed as a fine-tuning step exclusively for the native (finest) resolution. In this design we are exploiting the exponential map parameterization. Doing this fine-tuning would be harder to achieve without matrix exponential-based parameterization. Estimation of v^1 is still dependent on v, because even for the finest resolution v contributes most in constructing the transformation matrix. However, v and v^1 are independent variables.
>
> We will add these references in the final version.
>
> We can cut the optimization part a bit short to bring in some ablation studies from the appendix in he final version.
>
> When frameworks support GPU processing (for eg. AMI), we do utilize them. Currently it's the best case performance of all algorithms that are being compared.
>
> Our framework is general because matrix exponential can represent rigid-body/affine/perspective transformation. We have chosen to use affine transformation so that we would be able to compare with all the competitive methods.

---

> > ### Comment · AnonReviewer4 · 2020-04-03
> > **No change in initial score**
> >
> > I thank the authors for the response, which provides some clarifications, but does not address my initial major concerns.
> > Regarding v and v^1, I do not agree with the explanation. I understand that v and v^1 are independent variables, but v^1 is not just a fine-tuning: it can completely undo v (e.g. by the choice v^1= -v). It makes level 1 completely independent.

---

> > > ### Author Response · Authors · 2020-04-03
> > > **fine-tuning**
> > >
> > > v^1 can indeed undo some effect of v only on the final resolution. This is deliberate, because Gaussian scale-space shifts features (e.g., see "Scale-space theory in computer vision" by Tony Lindeberg, or many other classical work in space-space theory). Use of v^1 does not hurt results, but can only improve it, when possible.

---

### Meta-Review · Area_Chair1 · 2020-04-07
**MetaReview of Paper64 by AreaChair1**

**Rating:** 3
**Recommendation For Accepted Papers:** Poster

**Metareview:**

The paper presents a new image registration method. The method is developed based on mutual information (with MINE), matrix exponential for transformation matrix, and multi-resolution approach. Based on the reviews, replies and the paper, the proposed method is interesting and has been compared with different approaches. The replies from the authors have addressed most of the concerns (although not all) raised by the reviewers.

**Paper Type:**

both

**Special Issue:**

no

---

### Decision · Program_Chairs · 2020-04-11

Accept